# Recent Advances in Therapeutics for Severe Fever with Thrombocytopenia Syndrome Virus

**DOI:** 10.3390/v17091174

**Published:** 2025-08-28

**Authors:** Huimin Dang, Yuanyuan Wang, Lihong Zhang, Shan Xu, Lei Liu, Yigang Tong

**Affiliations:** 1School of Basic Medicine, Jiamusi University, Jiamusi 154007, China; 2Beijing Advanced Innovation Center for Soft Matter Science and Engineering, College of Life Science and Technology, Beijing University of Chemical Technology, Beijing 100029, China

**Keywords:** SFTSV, antiviral therapy, targeted therapy, combined therapy

## Abstract

Severe fever with thrombocytopenia syndrome virus (SFTSV) is a tick-borne bunyavirus with a mortality rate of up to 30%. There is no specific treatment for SFTSV. This article systematically reviews the progress of major anti-SFTSV drugs. The nucleotide analogues (favipiravir, 4′-fluorouridine diphosphate prodrug VV261) have shown clinical potential. Calcium channel blockers (nifedipine, etc.) block virus invasion by inhibiting calcium influx. Monoclonal antibody (S2A5/SNB02) has achieved targeted therapy, and SNB02 nanoantibody has entered clinical trials. However, many candidate agents predominantly focus on a single target, such as viral RdRp or host calcium channels, which makes it difficult to block the entire viral replication cycle and may accelerate the accumulation of resistant mutations. In addition, the low bioavailability of small-molecule drugs, the obstacles to industrial-scale production of antibody-based therapies, and the lack of Phase III clinical evidence severely restrict their clinical translation. Future research should focus on exploring viral replication mechanisms, developing drugs against key viral proteins, and designing multi-target combination therapies and novel drug delivery systems.

## 1. Introduction

Severe fever with thrombocytopenia syndrome (SFTS), caused by the severe fever with thrombocytopenia syndrome virus (SFTSV), is an acute febrile illness characterized by thrombocytopenia and leukopenia. In severe cases, it may trigger cytokine storms leading to multiple organ failure and death [1]. Clinical laboratory findings frequently demonstrate elevated serum levels of aspartate aminotransferase (AST), alanine aminotransferase (ALT), and lactate dehydrogenase (LDH) [2].

SFTSV, also designated *Dabie bandavirus*, belongs to the genus *Bandavirus* within the family *Phenuiviridae* and the order *Bunyavirales* [3,4]. The primary vector for transmission is the hard tick *Haemaphysalis longicornis* [5]. Exhibiting broad host tropism and zoonotic potential, SFTSV transmits not only via tick bites but also through direct blood exposure [6]. Since its initial identification in China in 2009 [2,7], the virus has spread to regions including South Korea, Japan, and Vietnam, with a global case fatality rate approximating 30% [8,9,10].

The absence of approved vaccines or targeted therapeutics poses a substantial threat to public health should large-scale outbreaks occur. Consequently, the WHO designated SFTSV as a priority pathogen in 2017 and reaffirmed its status as a high-risk agent under the Public Health Emergency of International Concern (PHEIC) framework in 2024. Elucidating the viral pathogenesis and developing effective countermeasures remain imperative.

## 2. Genomic Characteristics of SFTSV

SFTSV is a segmented, single-stranded, negative-sense RNA virus with spherical virions measuring 80–100 nm in diameter [2,11]. The viral genome comprises three distinct segments designated as small, medium, and large segments [12]. Structurally, each genomic segment is flanked by non-translated regions (NTRs) that form panhandle-like secondary structures at the 5’ and 3’ termini [13]. The large (L) segment has a full length of approximately 6368 base pairs (bp) and primarily encodes the viral RNA-dependent RNA polymerase (RdRp). This enzyme represents the primary target for nucleoside analogue-based antivirals [2,13]. The M segment (3378 bp) expresses a glycoprotein precursor processed into Gn and Gc glycoproteins—key targets for vaccine development. Gn facilitates cellular entry by binding the host receptor non-muscle myosin heavy chain IIA (NMMHC-IIA), while Gc mediates pH-dependent membrane fusion [14]. The S segment, approximately 1744 bp in length, encodes both the nonstructural protein (NS) and nucleoprotein (NP) [13]. NS antagonizes IFN-β production through TBK1 sequestration and impedes NF-κB signaling, processes that provoke dysregulated cytokine release and hyperinflammation. Concurrently, the oligomerization of NP into ring-like structures encapsulates the viral RNA-RdRp complex, assembling stable RNPs essential for genomic protection and viral replication. The open reading frame (ORF) within the S segment exhibits significantly lower conservation than its counterpart in the L segment [13]. Driven by genetic instability, NSs undergo frequent mutations that can mediate resistance to selected antiviral therapies. Hence, in the process of drug development, it is important to thoroughly account for the potential impact of viral genomic mutations, design NS-targeted drugs to counter mutation-driven resistance, and explore the Gn/Gc fusion mechanism as a dual-target inhibitor (Figure 1, Table 1).

## 3. Research Progress of SFTSV Antiviral Drugs

Current clinical management primarily involves supportive care (physical/pharmacological antipyresis, platelet transfusion, and glucocorticoid administration) and adjunctive herbal therapies (such as Yinqiao Powder, Xuebijing Injection, and Zhuye shigao Decoction) tailored to disease severity [15]. In addition, some broad-spectrum antiviral agents have not yet provided sufficient and reliable clinical evidence to prove their effectiveness [16]. Therefore, the development of effective anti-SFTSV drugs is an urgent need for current research.

### 3.1. Nucleotide Analogues

#### 3.1.1. Ribavirin

Ribavirin exerts broad-spectrum antiviral activity by disrupting mRNA capping, inhibiting RNA-dependent RNA polymerase (RdRp) function, and targeting IMP dehydrogenase [16,17,18]. Nevertheless, its clinical utility in SFTS management remains contentious due to inconsistent therapeutic outcomes.

Shimojima et al. performed anti-SFTSV experiments with ribavirin in Vero, Huh7, and U2OS cell lines, respectively, and demonstrated that ribavirin had an inhibitory effect on virus replication [19] (Table 2). Then, Lee et al. demonstrated that the drug could continuously inhibit viral replication within 48 h in an in vitro model [20]. However, the results of clinical studies have not proved that ribavirin treatment can significantly improve the prognosis of patients. Therapeutic efficacy assessment in 311 SFTS patients demonstrated no statistically significant difference in clinical fatality rates between the ribavirin treatment group (*n* = 138; 17.4%) and the non-ribavirin control group (*n* = 164; 17.1%) [21]. Furthermore, ribavirin administration showed no substantial impact on thrombocyte counts or viral load dynamics during the treatment course. The above results indicate that ribavirin may only have an antiviral effect in the early stage of infection and when the viral load is low (1 × 10^6^ copies/mL), but it cannot improve the survival rate of patients with high viral load [22]. Given the current constraints of available drugs, the development of more potent nucleotide analogues is urgently needed.

#### 3.1.2. Favipiravir

The pyrazinecarboxamide derivative favipiravir (6-fluoro-3-hydroxy-2-pyrazinecarboxamide, T-705), an approved broad-spectrum RNA viral inhibitor [23,24], functions through dual mechanisms of action (Figure 2): (1) eliciting lethal mutagenesis in RNA viruses, characterized by guanine-to-adenine and cytosine-to-thymidine transversions [25]; (2) competitively suppressing RdRp activity, thereby terminating viral RNA synthesis [26].

In the in vivo study conducted by Tani et al. on IFNAR1^−/−^ C57BL/6 mice, administration at 300 mg/kg resulted in stable body weight, significantly enhanced survival rate, and minimal viral RNA loads compared to ribavirin and placebo control groups [27]. Histopathological and immunohistochemical analyses confirmed preserved tissue integrity in cervical lymph nodes, liver, and kidneys, alongside attenuated inflammatory responses [27]. In vitro studies have confirmed that favipiravir is effective against multiple SFTSV strains, with an EC_50_ of 6.0 μM in Vero cells, significantly lower than that of ribavirin (40 μM) [28].

Li et al. performed a single-blind randomized controlled trial (*n* = 145), in which favipiravir treatment reduced the 28-day mortality to 9.5% among 74 patients, compared with 18.3% in the control group (*p* < 0.05). Subgroup analysis indicated that patients with a lower viral load (RT-PCR Ct ≥ 26) benefited most, with mortality declining from 11.5% to 1.6%, underscoring the significance of early therapeutic intervention [25]. In a separate single-arm study conducted by Suemori et al. (*n* = 23), the 28-day overall mortality was 17.3%. However, mortality reached 40% among patients with high viral loads (≥10^6^ copies/mL), while no deaths were observed in those with low viral loads (*p* = 0.013). Furthermore, the proportion of patients with positive virus isolation decreased from 78% to 12% after 7 days of treatment, suggesting potent viral clearance activity [29]. These results highlight favipiravir’s capacity for rapid viral clearance and mortality reduction, particularly in early-stage patients, though limitations such as small sample sizes and the absence of large-scale Phase III trials underscore the need for further clinical validation.

Taken together, favipiravir can substantially reduce mortality in SFTS, particularly in early-stage or low-viral-load patients, offering advantages over previous drugs such as ribavirin, with a manageable safety profile.

#### 3.1.3. 4′-Fluorouridine and Its Derivatives

Following favipiravir, researchers have been dedicated to developing novel nucleotide analogues with enhanced potency. By constructing a recombinant SFTSV carrying the Nanoluc luciferase reporter gene, Xu’s team achieved real-time visual monitoring and quantitative analysis of viral infection dynamics for the first time [30]. Based on this model, the research team identified that the nucleoside analogue 4′-Fluorouridine exhibits an EC_50_ of 5.499 μM in Vero cells and provided protection to A129 mice with a single oral dose of 20 mg/kg, but the compound exhibits poor chemical stability [30,31]. A double-prodrug structural optimization strategy was implemented by Cheng’s team, resulting in VV261—a derivative demonstrating exceptional antiviral potency (EC_50_ = 0.89 μM, CC_50_ > 100 μM) through enhanced cellular uptake and sustained release. A single oral dose exceeding 10 mg/kg achieved 100% survival in IFNAR1^−/−^ mice [32] (Table 2 and Table 3). VV261 has advanced to Phase I clinical trials in China, making it the first SFTS-targeted nucleoside analogue beyond favipiravir to progress into clinical development [32].

### 3.2. Calcium Channel Blockers

Targeting Cav1.2 channels, calcium channel blockers (CCBs) like benidipine and nifedipine exert antiviral activity through Ca^2+^ influx blockade(Figure 2). Critical validation stems from Li et al.’s findings: SFTSV-induced intracellular Ca^2+^ surges directly enhance viral RNA polymerase processivity, confirming calcium signaling as a druggable host factor for viral replication [33]. Nifedipine-mediated L-type calcium channel antagonism disrupts Ca^2+^-dependent viral membrane fusion, yielding potent anti-SFTSV activity in Vero cells (EC_50_ = 1.412 μM; CC_50_ = 96.92 μM) with a selectivity index of 68.6 [33]. Urata et al. not only delineated the antiviral activity profiles of manidipine— a representative CCB—in SW13 cells (EC_50_ = 2.83 μM, CC_50_ = 57.03 μM) and Huh7 cells (EC_50_ = 3.17 μM, CC_50_ = 28.2 μM) (Table 2), but also elucidated the underlying mechanism of action targeting viral replication machinery [34]. Research demonstrates that SFTSV exploits G-actin-mediated vesicular trafficking for cellular entry, whereas CCBs suppress calcium influx, thereby inhibiting the depolymerization of F-actin to G-actin and ultimately disrupting viral internalization and genomic replication [34]. In vivo experimental data demonstrate that CCBs significantly reduce viral loads and mortality rates in mice infected with SFTSV [33,34].

To evaluate the clinical efficacy of CCBs, Li et al. conducted a retrospective cohort study involving 2087 patients with SFTS, of whom 83 patients were treated with nifedipine before and during admission, 48 with non-nifedipine, and 249 untreated SFTS patients served as the reference group. The case fatality rate (CFR) in the nifedipine-treated cohort was 3.6%, significantly lower than that observed in both the reference SFTS group (19.7%) and the non-nifedipine antihypertensive therapy cohort (20.8%). The above results can be concluded that nifedipine can significantly reduce the mortality of SFTS patients, relieve clinical symptoms, and improve the prognosis of severe patients [33]. Therefore, regulating calcium influx to inhibit viral replication has emerged as a highly promising therapeutic strategy for SFTS.

### 3.3. Natural Products

#### 3.3.1. Vitamin D Derivatives

In the screening of natural products, in vitro assays using Huh7 cells demonstrated that doxercalciferol, alfacalcidol, and 1alpha-hydroxyvitamin D4 exhibit superior antiviral efficacy compared to the positive control agent favipiravir (EC_50_ = 11.97 ± 4.94 μM). Their respective EC_50_ values were 1.98 ± 1.30 μM, 1.59 ± 0.35 μM, and 2.72 ± 0.40 μM, with CC_50_ > 5 μM. In vivo, doxercalciferol and alfacalcidol improved survival in suckling mice and significantly reduced viral loads in six-week-old BALB/c mice. Notably, co-administration with favipiravir produced synergistic antiviral effects. Overall, vitamin D derivatives demonstrated promising activity in cell culture and provided partial protection in suckling and BALB/c mice, but failed to improve survival in adult IFNAR^−/−^ A129 mice [35]. At present, no clinical evidence in humans is available, and further studies are required before these compounds can be advanced as therapeutic candidates.

#### 3.3.2. Caffeic Acid

Caffeic acid (CA), a hydroxylated cinnamic acid derivative, exhibits antineoplastic and antimicrobial effects alongside potent antioxidant activity. It directly inhibits SFTSV entry by disrupting viral attachment to host cell receptors. Ogawa et al. reported that in in vitro tests using Huh7.5.1–8 cells, CA exhibited an EC_50_ of 48 μM and a CC_50_ of 7.6 mM [36] (Table 2). Mechanistic evidence confirms that CA exerts antiviral effects via allosteric inhibition of viral Gn/Gc glycoproteins—disrupting pH-dependent membrane fusion in acidic endosomes—thereby reducing SFTSV infectivity. Given its mechanism of action, CA primarily inhibits SFTSV during the early stages of infection [36,37]. The available data are confined to in vitro assays, and the absence of animal or clinical evidence underscores the need for additional research before CA can be considered a viable therapeutic candidate.

### 3.4. Amodiaquine

Amodiaquine is categorized as a chloroquine-like drug and is primarily used for antimalarial treatment. Baba et al. conducted a systematic screening against SFTSV and proposed that the compound and its derivatives may serve as selective inhibitors of SFTSV [28]. Subsequent in vitro assays in both Vero and Huh7 cell lines demonstrated that derivative C-90 exhibits potent inhibitory activity against SFTSV, with an EC_90_ of approximately 2.6 μM and a CC_50_ exceeding 50 μM. Animal model outcomes exhibited marked discrepancies from in vitro data: In C57BL/6J mice infected with 100 TCID_50_, mortality rates showed no significant reduction despite high-dose therapy (100 mg/kg for 5 consecutive days) [38] (Table 3). Pharmacokinetic analysis revealed subtherapeutic plasma drug concentrations in both 20 mg/kg and 100 mg/kg dose groups, indicating poor oral bioavailability [38]. Based on the aggregate clinical evidence, a cautious assessment of these agents’ therapeutic efficacy against SFTSV remains warranted for clinical management.

### 3.5. Monoclonal Antibodies

#### 3.5.1. Traditional Monoclonal Antibodies

Substantial progress has been achieved in developing monoclonal antibodies (mAbs) targeting the SFTSV surface glycoprotein Gn. Among seven mAbs engineered by Ren et al., S2A5, S1G3, and S1H7 inhibit multiple steps of viral infection—including viral attachment and membrane fusion—whereas B1G11 specifically blocks the attachment process [39]. Functional divergence of epitopes is directly reflected in neutralization potency: In SFTSV pseudotyped virus neutralization assays, S1G3, S2A5, and S1H7 exhibit EC_50_ values below 0.006 μg/mL, whereas B1G11 demonstrates an EC_50_ of 0.053 μg/mL [39]. Nevertheless, B1G11 manifests high efficacy against authentic viral strains of both QD02 (EC_50_ < 0.35 μg/mL) and WCH97 (EC_50_ < 0.05 μg/mL) genotypes. In the IFNAR1^−^/^−^; murine model (6–8 weeks, HBMC5 strain), a single 400 μg dose of S2A5 achieved 100% survival, positioning it among the most potent neutralizing monoclonal antibodies (Table 3). Subsequent structural analysis revealed the S2A5-bound Gn region exhibits high conservation during viral evolution, establishing a foundation for broad-spectrum therapeutics [39].

#### 3.5.2. Nanobody Technology

To overcome the limitations of conventional monoclonal antibodies—specifically their large molecular size and poor tissue penetration—Wu et al. employed camelid-derived nanobody technology. Through immunization of camelids with Gn protein and subsequent construction of a phage display library, 23 monoclonal antibodies were obtained, among which SNB02 exhibited optimal binding activity (EC_50_ = 1.05 μg/mL) [40] (Table 2). SNB02 effectively suppressed SFTSV infection in 4–6-week-old NCG-HuPBL humanized mice, demonstrating capabilities in vascular endothelial repair and attenuation of pathological injuries in pulmonary, hepatic, and renal tissues induced by viral pathogenesis [40]. In addition, Y-Clone Medical Science Co., Ltd., Jiangsu, China, and PanGen Biotech Inc., Suwon, Republic of Korea, established a limited technology transfer and collaborative R&D agreement for SNB02 on 10 January 2020, reflecting ongoing efforts to advance its development toward clinical applications [41].

### 3.6. Interferon and Its Inducers

#### 3.6.1. IFN-γ

IFN-γ is exclusively categorized as the type II interferon, mediating immune responses and exhibiting antiviral activity [42]. Studies demonstrate that during SFTSV infection, elevated systemic IFN-γ concentrations activate the JAK-STAT signaling cascade, facilitating the expression of antiviral genes [42,43] (Figure 2). Ning’s team demonstrated IFN-γ-induced suppression of SFTSV replication in HepG2 cells, with prophylactic administration (pre-infection) reducing viral titers by 2.7-log (TCID_50_ analysis) and post-infection intervention decreasing NP expression by 84.3% (immunofluorescence analysis) (Table 2) [44]. Moreover, the NS protein of SFTSV sequesters STAT1 within NS-induced inclusion bodies (IBs), thereby suppressing interferon signaling transduction. This mechanism may confer resistance to IFN-γ therapy. In animal studies, prophylactic administration of IFN-γ confers effective protection against lethal SFTSV infection in mice [44].

U.S. FDA-approved IFN-γ for chronic granulomatous disease reduced SFTSV-associated mortality, positioning this repurposed immunotherapeutic agent as a cornerstone for managing bunyaviral infections.

#### 3.6.2. Tilorone Dihydrochloride

Yang et al. screened 2572 FDA-approved drugs and identified tilorone dihydrochloride as a low-molecular-weight interferon (IFN) inducer with potent anti-SFTSV activity. In Huh7.5 cells, tilorone dihydrochloride exhibited an EC_50_ of 0.42 μM, demonstrating significantly greater potency than conventional antivirals [45] (Table 2). In ICR mice, treatment reduced neuroinflammation and improved survival to nearly 80%, while BALB/c mice displayed lower viral loads and diminished cytokine responses (Table 3). These findings support its role in stimulating host innate immunity. Yet, the lack of efficacy in IFNAR^−/−^ mice highlights its dependence on type I interferon signaling, raising questions about its use in immunocompromised settings. Moreover, reports of dose-related hepatotoxicity and potential immune imbalance remain major concerns. Overall, tilorone demonstrates encouraging preclinical activity, but its safety profile and mechanism-related limitations must be addressed before clinical translation can be realistically considered [45].

#### 3.6.3. Kaempferide

Kaempferol, a naturally occurring flavonoid, has been demonstrated by Du et al. to potentiate type I interferon (IFN) signaling [46]. Their study showed that this compound enhanced the antiviral activity of IFN-β against SFTSV at the cellular level, leading to further suppression of viral RNA synthesis and nucleoprotein (NP) expression. Notably, even in the absence of exogenous IFN, kaempferol monotherapy markedly suppressed SFTSV replication, an effect conceivably achieved through the augmentation of endogenous IFN responses (Figure 2).

Mechanistic studies revealed that kaempferide enhances and prolongs JAK/STAT activation triggered by IFN, facilitates the transcription of downstream effector genes, and suppresses SOCS3-mediated negative feedback. These actions collectively sustain phosphorylation of STAT1 and STAT2, leading to amplified antiviral immunity. Comparable effects were observed in cells infected with CCHFV, underscoring the broad-spectrum antiviral capacity of the compound [46].

These results position kaempferide among the limited number of natural small molecules with demonstrated direct activity against SFTSV in vitro, as well as broader efficacy against additional highly pathogenic viruses. However, considering that sustained IFN signaling could potentially exacerbate inflammatory reactions, further assessment in more physiologically relevant SFTSV models will be essential to fully evaluate its therapeutic potential.

#### 3.6.4. Metformin

A retrospective clinical analysis (*n* = 3225) identified acute hyperglycemia as an independent risk factor for fatal outcomes in patients with SFTS [47]. SFTSV infection triggers insulin resistance, leading to sustained hyperglycemia that amplifies inflammatory cascades. This metabolic imbalance initiates a vicious cycle through suppression of the AMPK–mTOR pathway: under high-glucose conditions, AMPK activity is downregulated, impairing autophagic flux and consequently accelerating both viral replication and disease progression [48].

In vitro, metformin was shown to inhibit SFTSV replication via activation of the AMPK–mTOR axis. Treatment with 620 μM metformin in Huh7 cells significantly reduced both SFTSV titers and nucleoprotein (NP) expression. In an in vivo model using C57BLKS/J-*Lepr^db^*/*Lepr^db^* mice (BKS-db/db) infected with SFTSV, metformin treatment at 200 mg/kg/d significantly reduced blood glucose levels from 30 mM to 15 mM and improved survival to 80%, compared to untreated controls (Table 3). Consistent with these results, clinical observations indicated that metformin-treated SFTS patients (*n* = 20) had significantly lower mortality compared to those receiving insulin monotherapy (*n* = 20) [48].

These collective findings argue that metformin exerts effects beyond glycemic control—modulation of autophagy indirectly suppresses SFTSV replication, highlighting its potential as an adjunct treatment in diabetic SFTS patients. Nevertheless, existing studies did not stratify outcomes by diabetes type (type 1 vs. type 2), and possible crosstalk with sex hormone pathways remains unclear. Future work should prioritize examining metformin–interferon combination therapy and developing targeted AMPK activators to strengthen antiviral potency.

### 3.7. Gn-Modified Biomimetic Nanospheres

RNA interference (RNAi) technology has emerged as a research focus for SFTSV therapy due to its sequence-specific gene silencing capability [49]. Small interfering RNA (siRNA) is susceptible to nuclease degradation and may induce immune responses, which hampers its clinical application. Wang et al. engineered SFTSV-targeted siRNA delivery systems by encapsulating nanoparticles with Gn-protein-modified Vero cell membranes, designated as Gn-modified biomimetic nanospheres, thereby overcoming key limitations of conventional siRNA carriers [50]. This nanoplatform not only enhances siRNA stability and reduces immunogenicity but also enables targeted delivery against SFTSV. In vitro assessments demonstrated potent antiviral efficacy in A549 and Vero cells (EC_50_ = 0.01844 μM), with CC_50_ exceeding 0.15 μM [50] (Table 2). Mechanistic studies demonstrate that this system silences the viral NSs protein gene and restores IFN-β expression levels [50] (Figure 2).

Compared to conventional pharmacological agents, siRNA-loaded nanoparticles achieve precise targeting through gene silencing mechanisms with enhanced safety profiles. Future investigations should validate in vivo antiviral efficacy, determine toxicity thresholds, assess targeting efficiency and pharmacokinetic properties, evaluate cross-neutralization capacity against SFTSV Gn protein mutants, and optimize long-term nanoparticle stability.

**Table 2 viruses-17-01174-t002:** Mechanisms of anti-SFTSV drugs and efficacy studies in cell models.

Drug	Cell Model	SFTSV Strain	EC_50/99_	CC_50_	Reference
Ribavirin	Vero	HB29	EC_99_ = 263 μM	>500 μM	[19]
	Huh7		EC_99_ = 424 μM		
	U2OS		EC_99_ = 78 μM		[28]
	Vero	Japanese isolate	EC_50_ = 40.1 ± 16.3 μM	>100 μM	
Favipiravir	Vero	SPL010	EC_50_ = 6.0 μM	>1000 μM	[27]
	Vero	Japanese isolate	EC_50_ = 25.0 ± 9.3 μM	>100 μM	[28]
VV261	Vero	HBMC16	EC_50_ = 0.89 μM	>100 μM	[32]
Manidipine	SW13 Huh-7	YG-1	EC_50_ = 2.83 μM IC_50_ = 3.17 μM	57.03 μM 28.2 μM	[34]
Doxercalciferol	Huh7	Chinese isolate	EC_50_ = 1.98 ± 1.30 μM	>9 μM	[35]
Alfacalcidol	Huh7	Chinese isolate	EC_50_ = 1.59 ± 0.35 μM	>9 μM	[35]
Caffeic acid	Huh7.5.1–8	HB29	EC_50_ = 48 μM	>100 μM	[37]
Amodiaquine	Vero Huh7	Japanese isolate	EC_50_ = 19.1 ± 5.1 μM	>100 μM	[38]
S2A5	Vero 293T	WCH97, QD02	EC_50_ ≤ 0.006 μM		[39]
SNB02	Vero	E-JS-2013-24	EC_50_ = 1.05 μM		[40]
IFN-γ	Vero	SPL030	EC_90_ = 0.012 μM		[44]
Tilorone dihydrochloride	Huh7.5	Chinese isolate	EC_50_ = 0.42 μM	10 μM	[45]
Kaempferide	293T	WCH-2011		>50 μM	[46]
Metformin	Huh7	HBMC16			[48]
Gn-modified biomimetic nanospheres	Vero	Chinese isolate	EC_50_ = 0.01844 μM	>0.15 μM	[50]

**Table 3 viruses-17-01174-t003:** In Vitro Efficacy of Anti-SFTSV Compounds in Cell Models.

Drug	Animal Model	Drug Dosage	Viral Inoculum	Survival Rate	Reference
Ribavirin	STAT2^−/−^hamster	75 mg/kg/d 25 mg/kg/d 60 mg/kg/d 300 mg/kg/d	50PFU		[19]
	IFNAR1^−/−^C57BL/6		10^6^TCID_50_		[27]
Favipiravir	IFNAR1^−/−^C57BL/6		10^6^TCID_50_		[27]
VV261	IFNAR1^−/−^C57BL/6	2.5, 5, 10 mg/kg/d	10^3^PFU	100%	[32]
Nifedipine	5W female C57BL/6	15 mg/kg/d	10^5^TCID_50_	100%	[33]
Manidipine	IFNAR1^−/−^C57BL/6	10 mg/kg/d	10PFU	50%	[34]
Doxercalciferol	ICR	0.3 μg/kg/d OR 0.15 μg/kg/d	1 × 10^5^PFU		[35]
Doxercalciferol Amodiaquine	6–8W A129(IFNAR1^−/−^)	10 μg/kg/d OR 5 μg/kg/d	10PFU		[35]
					[38]
	C57BL/6	100 mg/kg/d	100TCID_50_		
S2A5	6–8W IFN-α/βR	400 μg (6 hpi)	500TCID_50_	100%	[39]
SNB02	4–6W NCG-HuPBL	400 μg/kg/d	2 × 10^5^TCID_50_		[40]
IFN-γ	3d ICR	5 × 10^−4^ mg/kg	10^3^TCID_50_		[44]
Tilorone dihydrochloride	1d ICR	20 mg/kg	5 × 10^4^PFU	78.94%	[45]
Kaempferide	BALB/c	100 mg/kg/d	2 × 10^3^TCID_50_		[46]
Metformin	BKS^−^db/db	300 mg/kg/d	4 × 10^5^PFU	80%	[48]

## 4. Summary and Outlook

This review summarizes the recent advances in SFTSV antiviral drugs. At present, most of the drug research on SFTSV is still in the stage of in vitro experiments or animal models, and few have advanced into clinical trials. Nucleoside analogues like ribavirin and favipiravir are primarily effective early in infection or at low viral loads, though efficacy is often limited by emerging resistance. Calcium channel blockers are promising host-targeted drugs, with evidence from retrospective studies; however, their efficacy lacks confirmation from randomized controlled trials. Caffeic acid exhibits antiviral activity only at the cellular level, and its in vivo efficacy and safety remain to be fully evaluated. Despite demonstrated antiviral efficacy in vitro, the suboptimal oral bioavailability of amodiaquine substantially curtails its clinical translation potential. Vitamin D derivatives have demonstrated antiviral activity in both laboratory and animal studies and exhibit a synergistic effect when combined with favipiravir. However, this potential has not yet been validated in human subjects. The long-term administration of IFN-γ and its inducers necessitates rigorous risk assessment for therapy-limiting effects, including acquired resistance mediated by JAK-STAT pathway dysregulation and dose-dependent hepatotoxicity. Metabolic disturbances, particularly hyperglycemia, have been recognized as significant contributors to poor outcomes in SFTS. Early clinical observations indicate that metformin may help normalize glucose levels, lower viral burden, and correlate with improved survival. Yet, current evidence is based on small cohorts, and its therapeutic relevance in patients without hyperglycemia has not been established. But the SNB02 monoclonal nanobody demonstrates high efficacy in animal models and has advanced to preclinical and clinical investigations, representing one of the most promising translational strategies currently available.

Based on the above analysis, the majority of drug studies remain at the stage of in vitro experiments or animal models. In the future, it is imperative to conduct in-depth investigations into key viral proteins, such as NSs and Gn/Gc, as well as host factors, to facilitate the development of multi-target inhibitors. Moreover, advancing siRNA nanodelivery systems and self-assembling nanoparticle platforms must proceed in parallel, accelerating the clinical translation of vaccines through rigorous safety and efficacy evaluation, while establishing both SFTS rapid diagnostics and robust tick vector surveillance systems. These advances constitute pivotal scientific underpinnings for confronting SFTSV threats, forming the bedrock for sustainable safeguards of global health security.

## Figures and Tables

**Figure 1 viruses-17-01174-f001:**
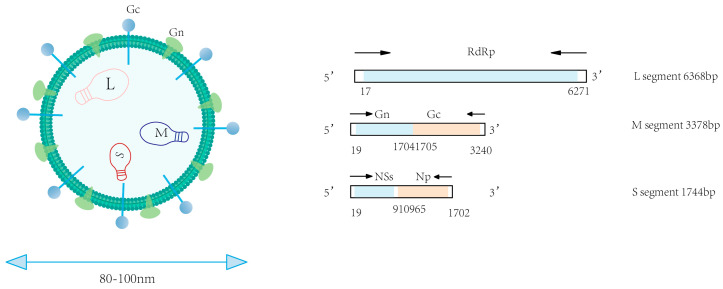
Genomic organization of SFTSV.

**Figure 2 viruses-17-01174-f002:**
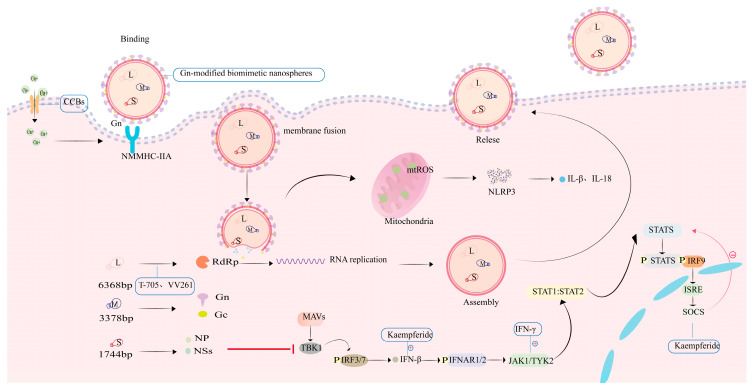
SFTSV replication cycle and pharmacological intervention targets. In the diagram, plus signs represent facilitating effects, minus signs represent inhibitory effects, and red arrows represent negative feedback inhibition.

**Table 1 viruses-17-01174-t001:** SFTSV genomic segments and encoded proteins.

Segment	Size	Encoded Protein	Function
L	6386 bp	RdRp	Catalyzes viral RNA synthesis (replication and transcription)
M	3378 bp	Gn	Mediates membrane fusion
		Gc	Coordinates viral attachment, entry, and membrane fusion
S	1744 bp	NSs	Induces cytokine storms and inflammatory responses
		Np	Assembles ribonucleoprotein complexes for genome protection

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
