# Peer review of "Recent Advances in Therapeutics for Severe Fever with Thrombocytopenia Syndrome Virus"

_viruses, 2025, doi:10.3390/v17091174_

Round 1
Reviewer 1 Report
Comments and Suggestions for Authors
The authors provide a detailed review of the current status of SFTSV drug development.The article is well written and does not raise any particular questions or issues. It is likely to attract the interest of many SFTSV drug researchers.
As a minor suggestion, the notation T-705 and favipiravir are mixed in the text, so I recommend using the generic name favipiravir throughout.
Reviewer 2 Report
Comments and Suggestions for Authors
The manuscript by Dang and colleagues addresses an important and timely topic, but requires considerable modification and editing to be of value. Most importantly, it is not at all a critical review, but simply presents brief summaries of studies on antiviral therapies for SFTSV infection. The authors need to summarize more explicitly which of these antivirals have actually been tested in human patients and whether they appeared efficacious. As it is written, I was not able to discern whether there are actually any antivirals that appear promising. As written, I judge this to require a major revision. One positive statement: the diagrams were very nice.
In addition to making this a critical review, there are numerous clarifications that need to be incorporated.
Lines 17-18: You state “At present, the main challenges are the single target and the difficulty of clinical transformation.” – I do not understand what that means.
Line 99-100: “Hence, antiviral drug development should fully consider the risk of resistance caused by the progressive accumulation of viral mutations” – this sentence seems to have no relevance to the discussion of ribavirin as an antiviral.
Line 101: You should add any reports in which favipiravir was used in human patients. Several such reports have been published.
Lines 128-129: “marking the first orally administered SFTS-targeted therapeutic candidate to reach the clinical stage” – this does not seem to be true – as favipiravir is also administered orally.
Line 160: Vitamin D Derivatives: again, please indicate whether or not these there are any reports of tehse compounds being used in anything except day-old mice, especially humans.
Line 169: State explicity whether Caffeic acid has been tested except in vitro.
Line 214: which animal models
Line 217: Need some reference that this antibody has entered human clinical trials (even a personal communication)
Line 238: again, please state what species when you say in vivo studies
Line 246+: Kaempferide and Metaformin: No indication that these have been tested with SFTSV (IAV [which you should define] is not relevant. Delete there sections if there is no information on SFTSV.
Minor comments:
Line 11: Delete novel – SFTSV is no longer a “new virus” and you could call almost any virus “novel”.
Line 108: delete “Hideki”
Line 152: Define CFR (clinical fatality rate?)
Throughout the manuscript, use past tense when reporting on previous studies (i.e. this antiviral “reduced” rather than “reduces” virus replication when citing a given report)
Comments on the Quality of English Language
Clearly needs copy editing for tense and spacing.
Round 2
Reviewer 2 Report
Comments and Suggestions for Authors
You have done an excellent job of transforming your manuscript into a valuable and critical review of the literature on therapeutics for SFTSV instead of simply listing numerous research reports without interpretation. Well done.